# Precision Landing Test and Simulation of the Agricultural UAV on Apron

**DOI:** 10.3390/s20123369

**Published:** 2020-06-14

**Authors:** Yangyang Guo, Jiaqian Guo, Chang Liu, Hongting Xiong, Lilong Chai, Dongjian He

**Affiliations:** 1College of Mechanical and Electronic Engineering, Northwest A&F University, Yangling 712100, China; yangyang.guo@uga.edu (Y.G.); guojiaqian@nwafu.edu.cn (J.G.); ripper_liu1016@163.com (C.L.); 18804636217@163.com (H.X.); 2Key Laboratory of Agricultural Internet of Things, Ministry of Agriculture and Rural Affairs, Yangling 712100, China; 3Shaanxi Key Laboratory of Agricultural Information Perception and Intelligent Service, Yangling 712100, China; 4Department of Poultry Science, College of Agricultural and Environmental Sciences, University of Georgia, Athens, GA 30602, USA

**Keywords:** unmanned aerial vehicle, mobile apron, precision landing, image detection, poultry house

## Abstract

Unmanned aerial vehicle (UAV) has been used to assist agricultural production. Precision landing control of UAV is critical for application of it in some specific areas such as greenhouses or livestock/poultry houses. For controlling UAV landing on a fixed or mobile apron/platform accurately, this study proposed an automatic method and tested it under three scenarios: (1) UAV landing at high operating altitude based on the GPS signal of the mobile apron; (2) UAV landing at low operating altitude based on the image recognition on the mobile apron; and (3) UAV landing progress control based on the fixed landing device and image detection to achieve a stable landing action. To verify the effectiveness of the proposed control method, apron at both stationary and mobile (e.g., 3 km/h moving speed) statuses were tested. Besides, a simulation was conducted for the UAV landing on a fixed apron by using a commercial poultry house as a model (135 L × 15 W × 3 H m). Results show that the average landing errors in high altitude and low altitude can be controlled within 6.78 cm and 13.29 cm, respectively. For the poultry house simulation, the landing errors were 6.22 ± 2.59 cm, 6.79 ± 3.26 cm, and 7.14 ± 2.41cm at the running speed of 2 km/h, 3 km/h, and 4 km/h, respectively. This study provides the basis for applying the UAV in agricultural facilities such as poultry or animal houses where requires a stricter landing control than open fields.

## 1. Introduction

The unmanned aerial vehicle (UAV) information acquisition technology has been applied in agricultural operations and it plays an important role in the development of precision agriculture [1,2,3]. UAV has been used for agricultural information acquisition [4,5,6,7,8], pesticide spraying [9,10,11], disease detection [12,13,14]. In recent years, the research of UAV has been focused on navigation and control direction. Among them, precision landing is a critical area in the field of UAV control study [15,16,17,18].

Visual navigation, as an effective navigation technology, receives increasing attentions. It mainly uses computer vision technology to obtain positioning parameter information [19,20]. This technology can obtain information through common low-cost devices and has broad application prospects in the field of UAV navigation. Sharp et al. [21] from the University of California at Berkeley studied the aircraft landing based on the computer vision system in 2001, and reported that the computer vision technology is applicable for UAV landing control. Anitha et al. [22] designed and realized automatic fixed-point landing of UAV based on visual navigation technology. Its fixed-point image had a well-defined geometric shape. After image processing, the image position was obtained, and the UAV’s flight attitude was adjusted to achieve landing. Sun and Hao [23] adjusted UAV autonomous landing based on the geometrical feature information of landmark, and the simulation under laboratory conditions showed that the method showed high stability and feasibility. Ye et al. [24] proposed a novel field scene recognition algorithm based on the combination of local pyramid feature and convolutional neural network (CNN) learning feature to identify the complex and autonomous landing scene for the low-small-slow UAV. For identifying the autonomous landing of an UAV on a stationary target, Sudevan et al. [25] used the speeded up robust features (SURF) method to detect and compute the key-point descriptors. The fast-approximate nearest neighbor search library (FLANN) was utilized to achieve the accurate matching of the template image with the captured images to determine the target position.

Most of the previous or existing UAV landing studies focused on a fixed location. In real agricultural operations (e.g., agricultural information collection and pesticide spraying), producers may like the UAVs to land on a mobile platform/apron (e.g., charging battery, data downloading, or refill pesticide) timely to improve the operational efficiency. In addition, in poultry or animal houses, UAVs may be used for monitoring indoor air quality and animal information. Precision control of landing on a mobile or a fixed apron is necessary. The objectives of this study were (1) designing a precision method for landing control of the UAV on a mobile apron automatically; (2) verifying the operations of the method under different scenarios (e.g., high altitude and low altitude); and simulating the operation of the UAV in the agricultural facility (i.e., a poultry house).

## 2. Materials and Methods

### 2.1. Hardware of Autonomous Precision Landing System

The system mainly consists of two parts: the UAV platform and the mobile apron (overall structure was shown in Figure 1). A four-rotor UAV was tested in the study. The UAV platform consisted of the following units: flight controller (Pixhawk 2.4.8) and its own GPS-1 unit; raspberry Pi 3b and its own camera (field angle: 72°; Sensor pixel: 1080p, 5 million pixels); wireless data transmission module-1 (WSN-02); relay (ZD-3FF-1Z-05VDC) and miniature electromagnet (ZS-20/15, DC: 12V), and the relay and the electromagnet formed an electromagnet circuit; the wireless data transmission module-1 received the GPS-2 signal from the apron and transmitted it to the flight controller; the flight controller combined its own GPS-1 signal information to obtain the relative position information of the two to adjust the flight attitude of the UAV. The camera obtained image and inputted it to the Raspberry Pi, and then the position information of the image label was acquired by the image recognition program loaded in the Raspberry Pi and transmitted to the flight controller. The flight controller would further adjust the flight attitude. The distance information acquired by the image recognition program was used to control the trigger electromagnet circuit.

The mobile apron was consisted of a wireless data transmission module-2 (WSN-02) and GPS-2 unit (NiRen-SIM868-V1.0). Iron plate had the diameter of 1 m and the thickness of 2 mm, and image label (35 cm × 35 cm). GPS-2 was connected with the wireless data transmission module-2 for emitting GPS-2 signals. Image tag was used for UAV image retrieval.

The electromagnet was embedded in the 3D printing die and connected with wire. When the electromagnet circuit was triggered and electrified, the electromagnet attracted the iron plate. Screws and nuts were used to fix the iron plate, and screws were embedded in the composite plate. Six screws and six nuts were selected and evenly distributed under the iron plate. Springs connected the iron plate and the composite plate acted as a shock absorber, and the spacing between the iron plate and the composite plate was about 2 cm.

### 2.2. UAV Landing Test Arrangement

Considering actual flight, experimental site and landing situation of the UAV, the test was conducted in three processes: (1) UAV landing at high operating ranges. The flight controller was used to adjust the UAV’s flight attitude by receiving and analyzing the GPS-2 signals of the apron and its own GPS-1 to approach the target; (2) UAV landing at low operating ranges. The image recognition method was used to detect the image label on the mobile apron and obtain the relative position information of the UAV and the apron for adjusting the flight parameters of the UAV to approach the mobile apron; and (3) UAV landing process. The fixed landing device was triggered according to image detection-based distance information to achieve a stable landing.

### 2.3. GPS Remote Positioning

When the UAV flied high and started to return (e.g., over 5 m from the apron), the GPS-2 signal of the apron would be received for coarse positioning. If the UAV was far away from the apron, the image label in the field-of-view would be small, so it would be not conducive to acquiring image features as the operation is easily interfered by the external environmental factors. Occasionally, the image label was not in the field-of-view. According to the longitude and latitude information transmitted by the GPS-2 signals of the apron and the longitude and latitude information of the UAV’s own GPS-1 signals, the relative position of the two can be quantified. Adjustment horizontal speeds of the UAV could be obtained according to the principle of triangular synthesis (Figure 2).

In order to avoid the situation that UAV and mobile apron may have the same horizontal speed and direction when UAV are not in the position vertical above the apron, the relative position information and relative speed information were integrated to adjust horizontal speed and descending speed of the UAV, as shown in Equations (1) and (2) [26,27].
(1)$$V_{F2}=V_{F1}+V_{F1}*\cos(\theta )$$
(2)$$V_{b0}=V_{h0}+V_{h0}*\sin(\theta )$$
where, *V_F2_* is the adjusted horizontal speed of the UAV. *V_F1_* is the horizontal speed of the UAV in the previous moment. *θ* is the angle between ΔP and UAV horizontal plane. ΔP is the actual distance vector from the UAV to the apron. *V_b0_* is the adjusted vertical speed of the UAV. *V_h0_* = 10 cm/s. *V_F2_* and *V_F1_* are vector values.

To improve the positioning accuracy, the longitude and latitude information obtained by the flight controller was continuously updated by the dynamic mean method. The schematic diagram was shown in Figure 3.

To reduce the GPS positioning deviation, the average values of 20 consecutive longitude and latitude information were used as the longitude and latitude inputs of the moving apron at the current moment, as described in Equation (3).
(3)$$P\left(x_{n},y_{n}\right)=\frac{1}{20}\sum_{i=n-19}^{n}G\left(x_{i},y_{i}\right),\;n=\{\begin{array}{c} 20,\;initial\;value\\ n=n+1,\;if\;i=n \end{array}$$
where, *x* is the latitude. *y* is the longitude. $P\left(x_{n},y_{n}\right)$ is the longitude and latitude finally obtained by flight controller. $G\left(x_{i},y_{i}\right)$ is the longitude and latitude detected in real time.

### 2.4. Accurate Positioning and Landing Based on Image Detection

When the height of UAV was about 5 m from the apron, the image recognition module would be turned on. If there was no image label in the field-of-view, the descending speed would be set to zero. The horizontal speed of the UAV was adjusted according to Equation (1) until the image label appeared in the field-of-view, then the flight attitude was adjusted to approach the apron based on the results of image detection/analysis. In Figure 1, the camera transmitted image to the Raspberry Pi, and the information extracted by the Raspberry Pi was transmitted to the flight controller for adjusting controlling parameters/inputs. According to the actual installation position of the camera, in the text, when the acquisition distance was less than 15 cm, the electromagnet circuit would be triggered.

#### 2.4.1. Camera Calibration

The camera equipped in UAV was calibrated for reducing distortion. In this study, 10 rows and 7 columns of chessboards were printed on an A4 paper, which was fixed on flat cardboard as the calibration board by following a published method [28].

Since the focal length of the camera was no longer changed after the camera was installed, it is necessary to measure the current field angle. According to the Equations (4) and (5), the horizontal field angle *θ*_1_ and the vertical field angle *θ*_2_ could be obtained.
(4)θ1=tan−1(d1h)
(5)θ2=tan−1(d2h)
where, *θ*_1_ is the horizontal field angle of camera. *θ*_2_ is the vertical field angle of camera. *d*_1_ is the actual distance from the center of left boundary to the center of the field-of-view. *d*_2_ is the actual distance from the center of upper boundary to the center of the field-of-view. *h* is the actual distance from the camera to the field-of-view.

#### 2.4.2. Image Label Detection

In this paper, the vision positioning system proposed by Edwin Olson [29] was used for image detection. To speed up the algorithm, the color image was transformed into gray image, and the image resolution was adjusted to 640 pixels × 480 pixels. The image label is composed of a large icon (35 cm × 35 cm) and a small icon (4 cm × 4 cm), as shown in Figure 4. When the UAV was close to the apron, the large icon could not be captured completely. At this time, the relative position information could be obtained by detecting the small icon. The horizontal speed of UAV was adjusted by Equation (1). Since the vertical speed of UAV could not be too fast when it is about the land, Equation (6) was used to adjust the vertical velocity speed [26,27].
(6)$$V_{b1}=V_{h1}+V_{h1}*\sin(\theta )$$
where, *θ* is the angle between Δ*P* and UAV horizontal plane. Δ*P* is the actual distance vector from the UAV to the apron. *V_b1_* is the adjusted vertical speed of the UAV. *V_h1_* = 5 cm/s.

#### 2.4.3. UAV Landing Fixation

UAV tended to roll over due to uneven tilting or sliding of the apron when it was landing. In order to solve this problem, a landing fixture based on an electromagnet principle was designed (see Figure 1). According to the actual size of UAV, a 3D printing mold was designed for placing the electromagnet. The apron was an iron plate. The distance between the UAV and the apron was obtained based on image detection for determining whether UAV was about to land. If yes, the electromagnet circuit would be triggered, and eventually to complete the landing safely.

### 2.5. Error Calculation

The error for using the label side length information as a reference standard can be estimated in Equation (7):(7)$$\delta =\left|D_{1}/D_{2}-S_{1}/S_{2}\right|/\left(S_{1}/S_{2}\right)$$
where, *δ* is the calculation error. *D*_1_ is the actual distance from the center of the label to the center of the field-of-view. *D*_2_ is the pixel distance from the center of the label to the center of the field-of-view. *S*_1_ is the actual side length of the label. *S*_2_ is the average of the four pixels side lengths of the label.

### 2.6. Poultry House UAV Landing Simulation

In poultry or livestock houses, environmental quality monitoring and animal imaging are important tasks for evaluating if animals have a good production environment and how their health and welfare are affected in commercial production system [30,31]. Traditional methods (e.g., installing sensors or cameras in fixed places) have limitations in selecting representative data collection locations due to the large size of the poultry house (e.g., 150 m length and 30 m wide) [32]. Therefore, equipping a UAV with different air quality sensors and cameras will promote the precision sensing of poultry housing environment and animal health/welfare in the future.

In this study, we simulated the UAV landing control by using the physical environments of a commercial poultry house as a model to evaluate effect of running speed on landing accuracy and how the size of an apron can be optimized for a precision landing control. The simulating house was measured 135 L × 15 W × 3 H m, as shown in Figure 5. We simulated that the UAV flew a round trip hourly and return back to a fixed location apron. In the simulation model, the image label was assumed as a large icon (35 cm × 35 cm) and a small icon (4 cm × 4 cm). The apron was set as 1 m high above floor in the end of the poultry house. Running speed of the UAV was set as 2 km/h, 3 km/h, and 4 km/h, respectively, to evaluate the effect of running speed on landing accuracy; the UAV running height was set as 2 m; the apron was assumed as a circle iron metal (1 m in diameter) at the height of 1 m above floor in the end of the house. The flying posture of the UAV could be adjusted automatically according to detected images label and *θ* (e.g., *θ* need to be ≥ 45° for control) based on the method in Equations (1) and (6).

## 3. Results and Discussion

### 3.1. Measured Field Angle

Figure 6 is a measurement diagram when the distance (*h*) between the camera and the wall is 313.6 cm. During the test, the center of the lens display (i.e., red UK-flag) was adjusted to make it coincide with the wall (the camera would be parallel to the wall, Figure 6). Black tapes were attached to the center of the left and upper edges to indicate the boundary of the field-of-view. On this basis, the actual distance *d*_1_ and *d*_2_ from the center of the left and upper boundaries to the center point of the UK-flag shaped were measured. In order to improve the measurement accuracy of the field angle, the measurement was performed at different distances *h*. The results were shown in Table 1 and Table 2. The measured horizontal and vertical field angles changed slightly at different distances *h*. The mean and standard deviation of the horizontal field angle were 25.18833° and 0.30308°, respectively, and the maximum and minimum differences between the measured horizontal field angle and the mean were 0.56142° and 0.04219°, respectively; The mean and standard deviation of the vertical field angle were 19.73093° and 0.28117°, respectively, and the maximum and minimum differences between the measured vertical field angle and the mean were 0.59921° and 0.06851°, respectively. Although the angle of view measured at different heights slightly changes, the change was negligible, so the average could be used as the true field angle of the camera. The image size of the camera is 640 pixels × 480 pixels (the ratio of width to length is 0.75). The average field-of-view ratio of the experimental measurement was 0.76273, and the maximum and minimum fluctuation values were 0.02111 and 0.00580, respectively.

### 3.2. Image Label Detection and Distance Model

#### 3.2.1. Image Label Detection

To verify the image detection method, images in different positions or rotations in the field-of-view were tested in the experiment, as shown in Figure 7a. In addition, anti-interference experiments were performed as shown in Figure 7b.

According to Figure 7a, the image detection method could detect the image label effectively even if the image label was in different positions (e.g., rotation). In addition, it can be seen from Figure 7b that the image detection method was highly resistant to interference. Therefore, this method is applicable in target detection during the test.

#### 3.2.2. Constructed Distance Model

In the test, the actual length of the edge of the image label was recorded as *S*_1_. The number of pixels on the same edge of the image label was recorded as *S*_2_. The actual distance corresponding to the pixel could be obtained by *S*_1_/*S*_2_, and this was used as a calculation standard. When the pixel distance from the center of the label to the center of the field-of-view was measured, the true distance could be obtained according to the calculation standard. In order to verify the feasibility of the calculation standard, the image labels with side lengths of 80 mm and 64 mm were tested at different distances *h*. As shown in Table 3 and Table 4.

Table 3 shows that the maximum and minimum calculation errors were 0.04 and 0.01, respectively. From Table 4, the maximum and minimum calculation errors were 0.06 and 0.01, respectively. *D*_1_/*D*_2_ could be represented by *S*_1_/*S*_2_, and be used as the reference standard to calculate the actual distance from the center of the label to the center of the field-of-view. In addition, the actual length of the edge of the image label was known, and the pixel value of the side length of the image label can be automatically calculated through the geometric relationship, so *S*_1_/*S*_2_ is feasible as a calculation standard.

Besides, Table 3 and Table 4 also show that the *S*_1_/*S*_2_ values at a different distance *h* were quite different, while the *S*_1_/*S*_2_ at the same distance *h* was relatively close. The actual distance from the center of the image label (80 mm × 80 mm as an example) to the center of the field-of-view was adjusted at different distances *h* to obtain a series of data (Figure 8).

As shown in Figure 8, the pixel distance would be in a linear relationship to the actual distance if the distance *h* from the camera to the wall was a constant. Then the *S*_1_/*S*_2_ value at this distance *h* can be calculated. The actual distance from the center of the image label to the center of the field-of-view can be calculated as the pixel value from the center of the image label to the center of the field-of-view multiplied by *S*_1_/*S*_2_. Briefly, the actual length of the whole field-of-view was obtained, and the height information was calculated by combining the field angle.

Therefore, through the above process, the flying height of the UAV was continuously and automatically obtained. After the height information was obtained, it would be judged if its value is greater than 5 m or not, then the flying speed of the UAV was adjusted according to either Equations (1) and (2) or Equations (1) and (6) until landed.

### 3.3. UAV Landing Accuracy

In order to evaluate the accuracy of the precision method for UAV landing control, different fixed-point and mobile landing experiments were conducted in an open area. The landing error (i.e., the actual horizontal distance from the center axis of the UAV to the center of the image label) was quantified. Figure 9 and Figure 10 shows the error distribution of fixed-point landing and mobile landing (apron moves at 3 km/h).

The average errors of fixed-point landing and mobile landing (3 km/h) were quantified as 6.78 cm and 13.29 cm, respectively, indicating that the newly developed method could control UAV landing on a mobile apron relative precisely. However, the landing errors were increasing with the increase of moving speed (>3 km/h) of the apron. This guarantee the further studies on optimization of control algorithms for UAV to land more precisely under different running speeds.

### 3.4. Simulation Results for the Poultry House

We simulated UAV landing control by using a real poultry house as a model to evaluate the effect of running speed on landing accuracy during applying the UAV in poultry house (e.g., monitor indoor environmental quality and animal health, welfare, or behaviors). It was estimated that a round trip of UAV would be 270 m in the poultry house. As the poultry house was 3 m high only (from ground to ceiling), so the flight height was recommended as 1.5–2.5 m for the UAV considering the facilities restrictions.

Table 5 shows the effect of UAV running speed on landing accuracy. The UAV running at a lower speed had a higher landing accuracy. In animal houses, a lower running speed is recommended considering the complex facilities and environment as compared to open field. The precision landing control provides a basis for optimize apron design in size to reduce the space use in the poultry house. This study is ongoing for studying the effect of the UAV weight on landing control efficiency to quantify how many sensors and cameras an UAV may carry in the future.

## 4. Conclusions

In the current study, a UAV landing control method was developed and tested for the agricultural application under three scenarios:(1)UAV landing at high operating altitude based on the GPS signal of the mobile apron;(2)UAV landing at low operating altitude based on the image recognition on the mobile apron;(3)UAV landing progress control based on the fixed landing device and image detection.

According to landing tests, the average errors of fixed-point landing and mobile landing at speed of 3 km/h were 6.78 cm and 13.29 cm, respectively. The UAV landing was smoothly and firmly controlled. We tested only limited parameters in the current study for the mobile apron landing (e.g., single speed at 3 km/h), further studies will be guaranteed to improve UAV landing control (e.g., detection, positioning, and control method of the mobile apron) by testing more parameters (e.g., speed and curve motion) in the future.

The simulation study for UAV application in agricultural facilities was conducted by using a commercial poultry house as a model (135 L × 15 W × 3 H m). In the simulation, the landing errors were controlled at 6.22 ± 2.59 cm, 6.79 ± 3.26 cm, and 7.14 ± 2.41 cm at the running speed of 2 km/h, 3 km/h, and 4 km/h, respectively. For the simulation work, we will continue to identify the effect of the UAV weight on landing control efficiency to promote the design of sensors and cameras in UAV for the real-time monitoring task in poultry houses. This study provides the technical basis for applying the UAV to collect data (e.g., air quality and animal information) in a poultry or animal house that requires more precise landing control than open fields. 

## Figures and Tables

**Figure 1 sensors-20-03369-f001:**
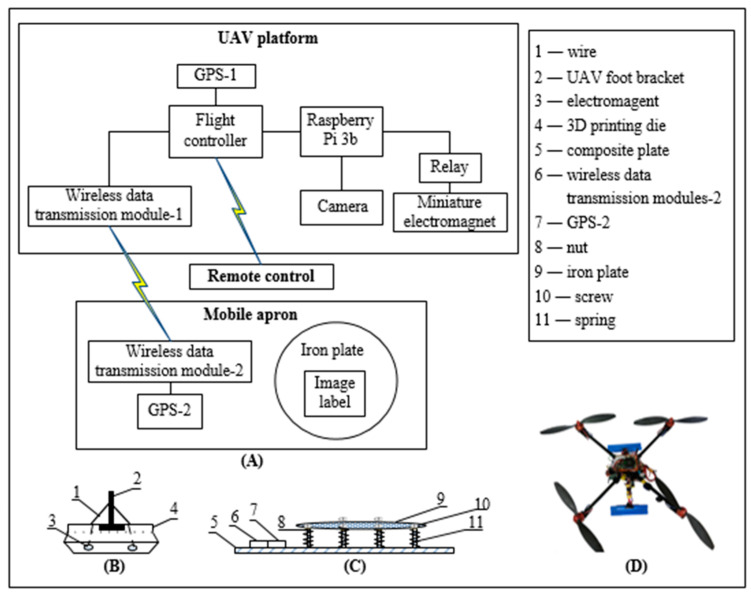
The schematic diagram of unmanned aerial vehicle (UAV) platform: (**A**) system overall hardware structure diagram; (**B**) schematic diagram of electromagnet installation structure; (**C**) mobile apron structural diagram; and (**D**) overhead view of the built UAV.

**Figure 2 sensors-20-03369-f002:**
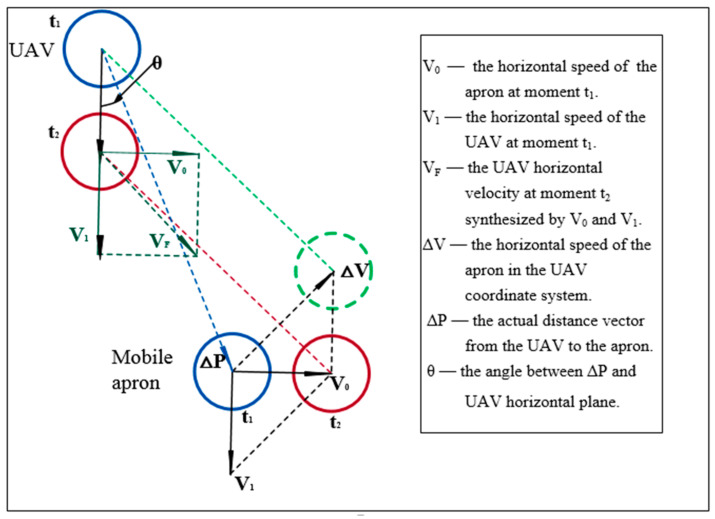
Triangular synthesis schematic diagram. The blue circle represents the UAV and the apron at time *t*_1_. The red circle represents the UAV and the apron at time *t*_2_, and the green dotted circle is the position of the apron in the UAV coordinate system. *V*_0_, *V*_1_, *V*_F_, ∆*V*, and ∆*P* are vector values, which contains both numerical values and directions. The horizontal speed of the UAV was synthesized by *V_F_* and ∆*V*.

**Figure 3 sensors-20-03369-f003:**
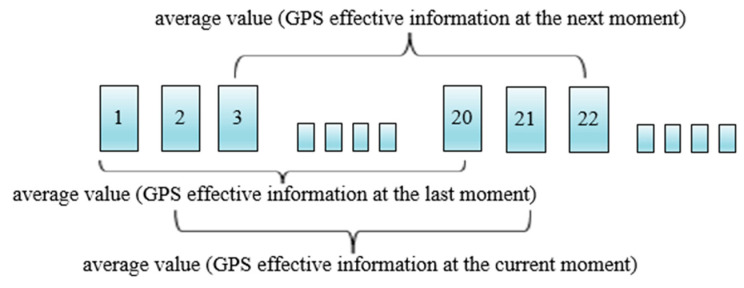
The principle of dynamic mean method (e.g., the 20th GPS information is the average of 1–20 GPS information).

**Figure 4 sensors-20-03369-f004:**
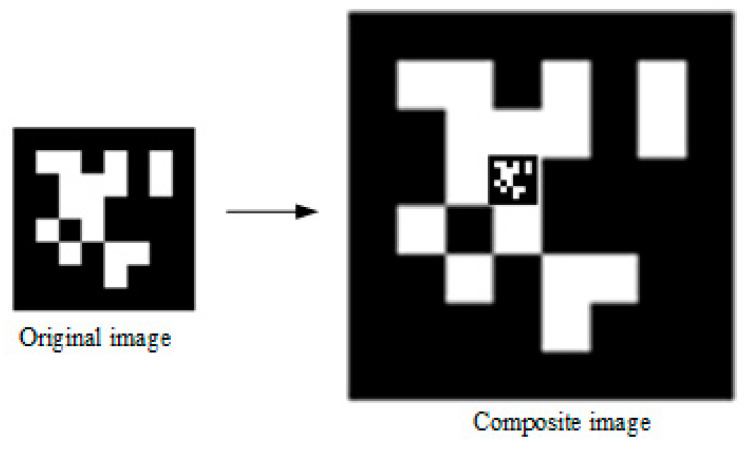
Image label. The image label is composed of a large icon (35 cm × 35 cm) and a small icon (4 cm × 4 cm).

**Figure 5 sensors-20-03369-f005:**
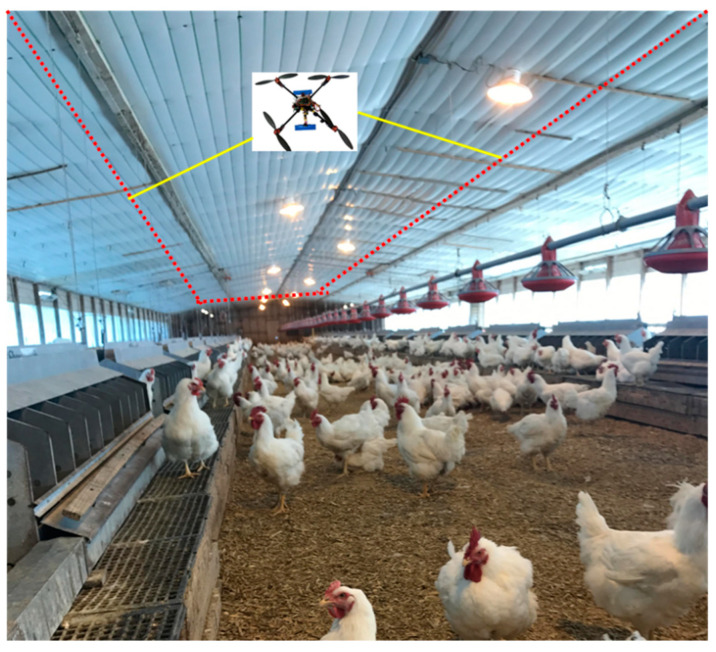
The poultry house model used for simulating UAV landing and control (the house was measured 135 L × 15 W × 3 H m; the photo was taken by authors in a commercial broiler breeder house in Georgia, USA).

**Figure 6 sensors-20-03369-f006:**
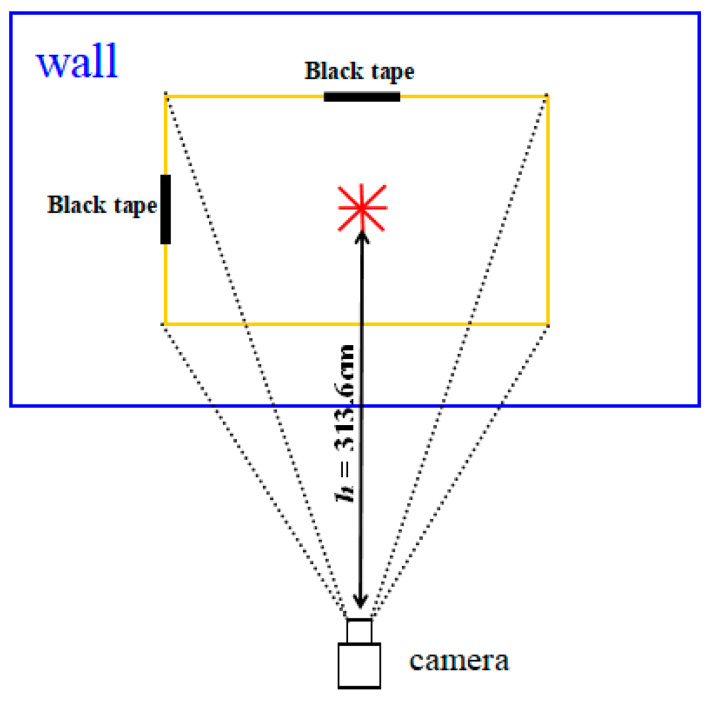
Measurement of the field angle. The blue frame is the wall, the yellow frame is the field of view of the camera, and *h* is the vertical distance from the camera to the center of the field of view. The center of the red UK-flag shaped is the center of the field-of-view, and black tapes are attached to the center of the upper and left boundaries of the field–of-view.

**Figure 7 sensors-20-03369-f007:**
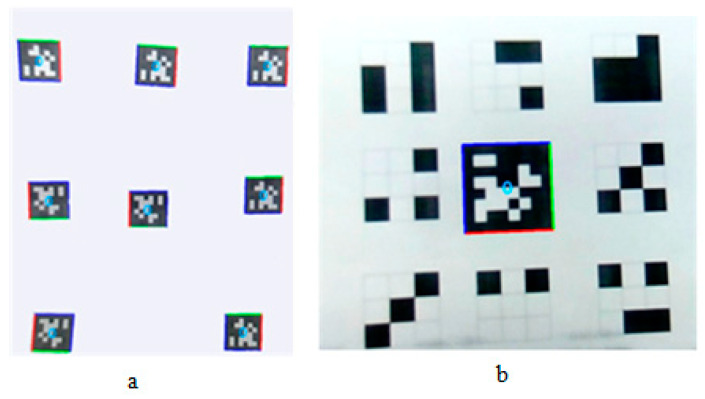
Image detection. (**a**). The detection results of images in different positions or rotations. (**b**). The detection results of anti-interference test.

**Figure 8 sensors-20-03369-f008:**
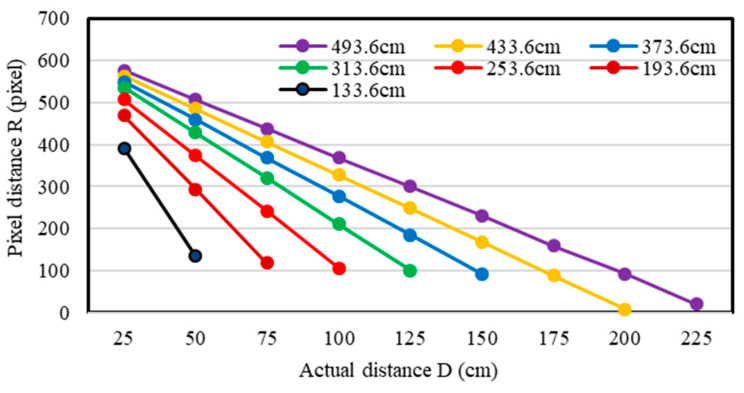
Distance relationship at different heights. The legend shows the actual distance from the camera to the wall. Horizontal coordinate is the actual distance from the image label to the center of the field-of-view. Vertical coordinates are the pixel distance from the image label to the center of the field-of-view.

**Figure 9 sensors-20-03369-f009:**
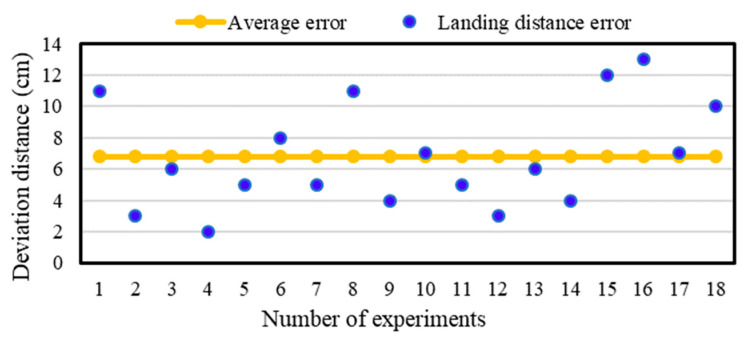
The error distribution of fixed-point landing. Horizontal axis is the number of experiments. Vertical axis is the distance from the center of the UAV landing position to the center of the image label. Blue dot is the distance error. Orange dot is the average error.

**Figure 10 sensors-20-03369-f010:**
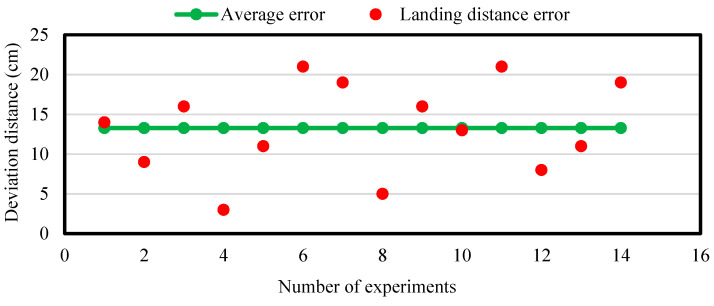
The error distribution of landing when the apron moves at 3 km/h. Horizontal axis is the number of experiments. Vertical axis is the distance from the center of the UAV landing position to the center of the image label. Red dot is the distance error. Green dot is the average error.

**Table 1 sensors-20-03369-t001:** Measurement results of field angle at different distances *h.*

Number	Distance *h* (cm)	Length *d*_1_ (cm)	Width *d*_2_ (cm)	*d*_2_/*d*_1_	Horizontal Field Angle *θ*_1_ (°)	Vertical Field Angle *θ*_2_ (°)
1	73.60	35.50	26.60	0.74930	25.74975	19.87053
2	133.60	62.60	49.50	0.79073	25.10607	20.33014
3	193.60	92.50	68.60	0.74162	25.53799	19.51123
4	253.60	119.50	91.60	0.76653	25.23052	19.85967
5	313.60	147.20	110.90	0.75340	25.14481	19.47540
6	373.60	172.00	134.50	0.78198	24.72068	19.79944
7	433.60	202.00	152.90	0.75693	24.97929	19.42415
8	493.60	230.50	175.50	0.76139	25.03153	19.57291
Mean ± SD				0.76273 ± 0.01548	25.18833 ± 0.30308	19.73093 ± 0.28117

**Table 2 sensors-20-03369-t002:** Variation of the field-of-view parameters.

Difference between *d*_2_/*d*_1_ and the Mean	Difference between *θ*_1_ and the Mean (°)	Difference between *θ*_2_ and the Mean (°)
−0.01343	0.56142	0.13960
0.02800	−0.08226	0.59921
−0.02111	0.34966	−0.21970
0.00380	0.04219	0.12874
−0.00933	−0.04352	−0.25553
0.01925	−0.46765	0.06851
−0.00580	−0.20904	−0.30678
−0.00134	−0.15680	−0.15802

**Table 3 sensors-20-03369-t003:** Distance test results of the image label (80 mm × 80 mm).

**Distance *h* (mm)**	**Distance from Label Center to Field-of-View Center**		
	**Real Distance *D*_1_ (mm)**	**Pixel Distance *D*_2_ (pixel)**	***D*_1_/*D*_2_ (mm/pixel)**
736.00	141.40	269.00	0.53
1936.00	707.10	482.00	1.47
1936.00	500.00	341.00	1.47
2536.00	707.10	369.00	1.92
2536.00	500.00	271.00	1.85
3136.00	707.10	297.00	2.38
3136.00	500.00	209.00	2.39
**Label Edge Length**			
**True Side Length *S*_1_ (mm)**	**Average Pixel Length *S*_2_ (pixel)**	***S*_1_/*S*_2_ (mm/pixel)**	**Error δ (mm/pixel)**
80.00	153.00	0.52	0.01
80.00	55.00	1.45	0.01
80.00	55.00	1.45	0.01
80.00	42.00	1.90	0.01
80.00	44.00	1.82	0.01
80.00	35.00	2.29	0.04
80.00	34.50	2.32	0.03

**Table 4 sensors-20-03369-t004:** Distance test results of image label (64 mm × 64 mm).

**Distance (mm)**	**Distance from Label Center to Field-of-View Center**		
	**Real Distance *D*_1_ (mm)**	**Pixel Distance *D*_2_ (pixel)**	***D*_1_/*D*_2_ (mm/pixel)**
736.00	145.00	286.00	0.51
736.00	191.40	357.00	0.54
1936.00	707.10	486.00	1.45
1936.00	500.00	374.00	1.34
2536.00	707.10	369.00	1.92
2536.00	500.00	285.00	1.75
3136.00	707.10	297.00	2.38
3136.00	500.00	232.00	2.16
**Label Edge Length**			
**True Side Length *S*_1_ (mm)**	**Average Pixel Length *S*_2_ (pixel)**	***S*_1_/*S*_2_ (mm/pixel)**	**Error δ (mm/pixel)**
64.00	124.50	0.51	0.01
64.00	120.00	0.53	0.01
64.00	46.50	1.38	0.06
64.00	47.00	1.36	0.02
64.00	34.00	1.88	0.02
64.00	36.00	1.78	0.01
64.00	28.00	2.29	0.04
64.00	28.50	2.25	0.04

**Table 5 sensors-20-03369-t005:** Effect of UAV running speed on landing accuracy (simulation times *n* = 15 times).

**Horizontal Running Speed**	2 km/h	3 km/h	4 km/h
**Mean ± SD**	6.22 ± 2.59 cm	6.79 ± 3.26 cm	7.14 ± 2.41 cm

Note: Vertical initial speed and adjustments can be found in Equation (6).

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
