# Peer review of "Precision Landing Test and Simulation of the Agricultural UAV on Apron"

_sensors, 2020, doi:10.3390/s20123369_

Round 1

Reviewer 1 Report

This is interesting and timely manuscript, which can promote even more use of UAV devices in agriculture. The article is also relatively well written, nevertheless, I found some inaccuracies.

Here are my comments:

Introduction section is OK, I think. Its focus corresponds to the presented results.

  1. 74 – It is unusual when after the end of one parenthesis starts second one.

Figure 3 is in bad quality.

  1. 186 – Abbreviation UVA should be changed to UAV.

The main problem I have with Results section. It starts at l. 207, where is the discussion about angles fluctuations. There are no units of this expression, degrees probably? The same l. 208 and 209.

  1. 210 – Please split theration to the ratio.

Units are missing also in Table 2 in all columns.

The main comment to this article from my side is to the results in Table 3. Here is measured distance in mm and pixel distance in pixels. The last column then shows both units in the proportion, again without unit. But there should be a unit of mm/pixel? Is it possible to compare these two units? The physical unit of a pixel is mm2, it is essentially a square. Is the resulting unit mm-1 then? Please explain it better in discussion.

There is something wrong in Fig. 7. One colour is not visible, and legend is for 6 dependencies only.

  1. 254 - The dot after sentence is missing.

In fig. 8 legend is missing to me.

Author Response

This is interesting and timely manuscript, which can promote even more use of UAV devices in agriculture. The article is also relatively well written, nevertheless, I found some inaccuracies.

Here are my comments:

Introduction section is OK, I think. Its focus corresponds to the presented results.

  1. 74 – It is unusual when after the end of one parenthesis starts second one.

Reply:  Thank you for the suggestion. We removed the second parenthesis and added “and” in line 75.

  1. Figure 3 is in bad quality.

Reply:  Figure 3 was modified as suggested.

  1. 186 – Abbreviation UVA should be changed to UAV.

Reply:  The “UVA” was corrected as “UAV”.

  1. The main problem I have with Results section. It starts at 207, where is the discussion about angles fluctuations. There are no units of this expression, degrees probably? The same 208 and 209.

Reply:  Thanks for the question. We modified the wring as suggested in Lines 207-216: “The mean and standard deviation of the horizontal field angle were 25.18833° and 0.30308°, respectively, and the maximum and minimum differences between the measured horizontal field angle and the mean were 0.56142° and 0.04219°, respectively; The mean and standard deviation of the vertical field angle were 19.73093° and 0.28117°, respectively, and the maximum and minimum differences between the measured vertical field angle and the mean were 0.59921° and 0.06851°, respectively. Although the angle of view measured at different heights slightly changes, the change was negligible, so the average could be used as the true field angle of the camera.”

  1. 210 – Please split theration to the ratio. Units are missing also in Table 2 in all columns.

Reply:  We split theration to the ratio and added units in Table 2 as suggested.

  1. The main comment to this article from my side is to the results in Table 3. Here is measured distance in mm and pixel distance in pixels. The last column then shows both units in the proportion, again without unit. But there should be a unit of mm/pixel? Is it possible to compare these two units? The physical unit of a pixel is mm2, it is essentially a square. Is the resulting unit mm-1 then? Please explain it better in discussion.

Reply:  Thanks for the suggestion. We added the unit of mm/pixel in Table 3. Besides, the following modifications were made to discussion: “According to Table 3, the maximum and minimum calculation errors were 0.046 and 0.01, respectively. Table 4 shows that the maximum and minimum calculation errors were 0.06 and 0.01, respectively. Therefore, D1/D2 could be represented by S1/S2, which could be used as a reference standard to calculate the actual distance from the center of the label to the center of the field-of-view. In addition, the actual length of the edge of the image label was known. The pixel value of the side length of the image label can be automatically calculated through the geometric relationship, so S1/S2 is feasible as a calculation standard.”.  “In Table 3 and Table 4, the S1/S2 values at different distance h were quite different, while the S1/S2 at the same distance h was relatively close.” Lines 250-261.

In addition, further explanation on two units were added to Lines 259-264: “As shown in Figure 7, the pixel distance would be in linear relationship to the actual distance if the distance h from the camera to the wall was a constant. Then the S1/S2 value at this distance h can be calculated. The actual distance from the center of the image label to the center of the field-of-view can be calculated as the pixel value from the center of the image label to the center of the field-of-view multiplied by S1/S2. Briefly, the actual length of the whole field-of-view was obtained, and the height information was calculated by combining the field angle.”

  1. There is something wrong in Fig. 7. One colour is not visible, and legend is for 6 dependencies only.

Reply:  Figure 7 was edited as suggested.

  1. 254 - The dot after sentence is missing.

Reply:  Corrected as suggested.

  1. In fig. 8 legend is missing to me.

Reply:  Figure 8 and Figure 9 were edited as suggested.

Reviewer 2 Report

The comments are in the file.

Author Response

All suggestions/comments made in PDF file were addressed.

  1. 16,34 – avoid using adjectives, “widely”.

Reply:  Thank you for the suggestion. We removed “widely” in lines 16, 35.

  1. 36 – document (4) out of context.

Reply:  The document (4) was removed as suggested.

  1. 37 – do not use etc.

Reply:  The “etc” was removed ass suggested, as well as “abundant” in line 43.

  1. 39 – document (19) from 2005 is not very recent.

Reply:  The reference was updated in lines 376-379. The citation # was changed from 37 8 in line 372.

  1. 50 – algorithm or method? had?

Reply:  The “algorithm” was changed into “method” as suggested. The “showed high stability and feasibility” was added. line 51.

  1. 50 – document (25) not found.

Reply:  This is a paper in the Chinese Journal of Computer Applications. You can find this article at the link below.

https://xueshu.baidu.com/usercenter/paper/show?paperid=09232820f5808c81182e1ec2667f57c3&site=xueshu_se&hitarticle=1

  1. 75 ,76, 85 – GPS-1 or GPS-2?

Reply:  Thanks for the question. The “GPS” was corrected as “GPS-2” in line 76, as “GPS-1” in line 77, and as “GPS-2” in line 86, respectively.

  1. 87 – organize the figures and give more resolution, labels (A), ... (C) can have a smaller dimension. 88 – changed “platform” to “apron”.

Reply:  Thanks for the suggestion. The Figure 1 was modified by changing “platform” to “apron”. Lines 89,90.

9.91 – changed “absorbed” to “attracted”

Reply:  The “absorbed” was changed as “attracted”. Line 93.

  1. 98 – changed “high altitude” to “high operating ranges”. 99 – changed “GPS” to “GPS-2”. 100 – changed “low altitude” to “low operating ranges”.

Reply:  Thanks for the suggestion. The “high altitude” was changed as “high operating ranges” in lines 20, 100, and 303; the “GPS” was changed as “GPS-2” in line 101; “low altitude” was changed as “low operating ranges” in lines 22,101, and 305; and “GPS-1” was added in line 102.

  1. 104 – removed “All tests were conducted at the Northwest A&F 104 University, Yangling, Shaanxi, China.”

Reply:  The sentence was removed as suggested.

  1. 107, 112, 113 – GPS-1 or GPS-2?

Reply:  We changed “GPS” to “GPS-2” in line 109, “GPS” to “GPS-2” in line 114, “GPS” to “GPS-1” in line 115.

  1. 116 – figure 2. It is suggested to improve the graph to reduce the amount of information in the title of the figure.

Reply:  Figure 2 was modified as suggested.

  1. 136 – it is suggested to improve the resolution of the graph.

Reply:  Figure was improved as suggested.

  1. 188 – removed “attracted the apron”

Reply:  It was removed as suggested.

  1. 218 – Table 1, ºC?

Reply:  We corrected “ºC” as “°” across the writing.

  1. 224 – Re-edit the figure 6.

Reply:  Figure 6 was edited as suggested.

  1. 254 - The dot after sentence is missing.

Reply:  Added as suggested. line 276.

  1. 259 - Please pay attention to point.

Reply:  We removed extra point as suggested. line 283.

  1. 264, 268 – figure8 and 9 should use labels as in figure 7.

Reply:  Figure 8 and Figure 9 were modified as suggested.

  1. 266 – Orange? the color that is perceived is yellow.

Reply:  Yes, it is orange.

  1. 274 – changed “>3 m/h” to “>3 km/h”.

Reply:  Corrected as suggested. Line 299.

Reviewer 3 Report

The manuscript addresses an interesting issue of UAV control, that is, autonomous landing onto a mobile apron. After reading carefully, here are my comments:

  1. The problem this paper is dealing with is not new and has been investigated in many existing papers. The solutions proposed in this manuscript is also not novel.
  2. The paper title and the abstract do not reflect correctly its main contents. The title is quite misleading as it mentions “landing control” and “agricultural UAV” while neither control problem nor control algorithm is proposed and no agricultural UAV (with some appropriate applications) is used in the work presented throughout the paper.
  3. The paper is written not very well and should be reconstructed to convey its contents to readers.
  4. The experiment data are provided quite detailed in tables. However, because there are many tables with numbers, the authors may want to discuss the table data more deeply to emphasize their contributions.

Author Response

Comments and Suggestions for Authors

The manuscript addresses an interesting issue of UAV control, that is, autonomous landing onto a mobile apron. After reading carefully, here are my comments:

  1. The problem this paper is dealing with is not new and has been investigated in many existing papers. The solutions proposed in this manuscript is also not novel.

Reply:  Thanks for the comments. UAV has been well studied in past decades, but there are still many knowledge gaps in this area such as how to apply the technology to solve problems in confined environmental instead of open area or crop field. In poultry or animal house, for instance, the UAVs need to return back to a precise location for charging, data backup, and other operations. Therefore, this study aimed to test and enhance landing accuracy on both fixed and mobile aprons. The novelty of this study as compared to others was highlighted in Lines 59-66.

Besides, this research provided technical thought for agricultural UVAs in landing control. The objectives of this study were designing a method for controlling the UAV landing precisely on mobile apron automatically under different scenarios (e.g., high altitude and low altitude). In addition, we designed a new landing fixture to achieve a safe landing. To improve the positioning accuracy, the longitude and latitude information obtained by the flight controller was continuously updated by the dynamic mean method. Related information is listed in lines 289-302.

  1. The paper title and the abstract do not reflect correctly its main contents. The title is quite misleading as it mentions “landing control” and “agricultural UAV” while neither control problem nor control algorithm is proposed and no agricultural UAV (with some appropriate applications) is used in the work presented throughout the paper.

Reply:  Thanks for the comments. Our landing control methods are introduced in Materials and Method in many places:

(1) when the UAV was landing at a high operating range, adjustment speeds of the UAV would be obtained according to Eqs. (1) and (2).

(2) when the UAV was landing at low operating ranges, the position information of image label on moving apron can be acquired based on image detection technology to adjust UAV flight attitude. The horizontal speed of UAV was adjusted by Eq. (1). Since the vertical speed of UAV couldn’t be too fast when it is about the land, Eq. (6) was used to adjust the vertical velocity speed.

(3) when the UAV was about to land. UAV tended to roll over due to uneven tilting or sliding of the apron when it was landing. In order to solve this problem, we designed a landing fixture based on electromagnet principle. when the UAV was about to land, the landing fixture was activated to achieve a safe landing.

In addition, following information was added clarify control principle: “Therefore, through the above process, the flying height of the UAV could be continuously and automatically obtained. After the height information was obtained, it would be judged if the value is greater than 5 m, then the flying speed of the UAV would be adjusted according to to Eqs. (1) and (2) or Eqs. (1) and (6) until landing” Lines 261-269.

  1. The paper is written not very well and should be reconstructed to convey its contents to readers.

Reply:  Thank you for the suggestions to improve the structure of the paper. As suggested, we adjusted writing in many places for improving the general structure.

(1) Objectives were refined: Lines 59-66.

(2) All figures (1-7) were adjusted to improve the structure or information for readers’ convenience.

(3) Tables were adjusted or further explained/discussed to provide readers more details. For instance. More information was added to explain table 3 and table 4, two most important results in Lines 250-261: “According to Table 3, the maximum and minimum calculation errors were 0.046 and 0.01, respectively. Table 4 shows that the maximum and minimum calculation errors were 0.06 and 0.01, respectively. Therefore, D1/D2 could be represented by S1/S2, which could be used as a reference standard to calculate the actual distance from the center of the label to the center of the field-of-view. In addition, the actual length of the edge of the image label was known. The pixel value of the side length of the image label can be automatically calculated through the geometric relationship, so S1/S2 is feasible as a calculation standard.”. “In Table 3 and Table 4, the S1/S2 values at different distance h were quite different, while the S1/S2 at the same distance h was relatively close.”

(4) Conclusions were refined. Lines 288-301.

  1. The experiment data are provided quite detailed in tables. However, because there are many tables with numbers, the authors may want to discuss the table data more deeply to emphasize their contributions.

Reply: Thanks for the suggestions/comments. We provided raw data or analyzed information in tables so readers can learn more details or compare their findings to our works more easily. Specially, more information was added to discuss two important tables (e.g., Tables 3 and 4): “Table 3 shows that the maximum and minimum calculation errors were 0.04 and 0.01, respectively. From Table 4, the maximum and minimum calculation errors were 0.06 and 0.01, respectively. D1/D2 could be represented by S1/S2, and be used as the reference standard to calculate the actual distance from the center of the label to the center of the field-of-view. In addition, the actual length of the edge of the image label was known, and the pixel value of the side length of the image label can be automatically calculated through the geometric relationship, so S1/S2 is feasible as a calculation standard. Besides, Table 3 and Table 4 also show that the S1/S2 values at different distance h were quite different, while the S1/S2 at the same distance h was relatively close.” Lines 240-248.

Round 2

Reviewer 2 Report

It point of the line 263 (s1/s2.) isn't visible.

Author Response

  1. It point of the line 263 (s1/s2.) isn't visible.

Reply:  Thank you for the suggestion. We added the missed point (period). Line 286. 

Reviewer 3 Report

The authors revised the manuscript based on the reviewers’ comments. After reading the updated version, here are my comments:

  1. The paper has some specific contributions to the body of knowledge such as the method to achieve highly accurate vision-based landing target position determination. 
  2. The authors said that “landing control methods are introduced in Materials and Method in many places” including Eqs (1), (2), and (6). However, there are some main issues here:
  • The equations are presented without an explanation and/or proof that allows readers, like me, to believe that they are correct and to understand how they were obtained.
  • These equations can be, somehow, called a feed-forward control method because they do not take the state errors (between the commands and the actual states) into account. However, in my opinion, these equations should not be considered as a control algorithm because they do not generate any control input that directly drives the vehicle.
  • It is required to prove the system stability when it comes to the control algorithm. However, no system stability analysis is presented in the work.
  1. Furthermore, once the title and abstract state about agriculture UAV, it will be supposed to show in the paper a UAV with some particular agricultural applications (that made it names agriculture UAV). However, neither simulation nor experiment related to such an UAV is showed. The figures for landing accuracy in subsection 3.3 are not enough to demonstrate the feasibility and effectiveness of the proposed method.
  2. Generally, the paper is written not very well in its previous version and it still seems to me that this version is not significantly improved. The way the authors present their methods, contributions, and demonstrations are all not clear and not convinced.

Author Response

Thank you very much for the suggestions/comments. We went through and addressed each of that. 

  1. The paper has some specific contributions to the body of knowledge such as the method to achieve highly accurate vision-based landing target position determination.

Reply: Thank you for the nice comments.

  1. The authors said that “landing control methods are introduced in Materials and Method in many places” including Eqs (1), (2), and (6). However, there are some main issues here:

The equations are presented without an explanation and/or proof that allows readers, like me, to believe that they are correct and to understand how they were obtained.

These equations can be, somehow, called a feed-forward control method because they do not take the state errors (between the commands and the actual states) into account. However, in my opinion, these equations should not be considered as a control algorithm because they do not generate any control input that directly drives the vehicle.

It is required to prove the system stability when it comes to the control algorithm. However, no system stability analysis is presented in the work.

Reply:  Thank you for the comments. We added a number of references to support our methods (e.g, references [26,27].  

“[26] Hu, M.; Li, C.T. Application in real-time planning of UAV based on velocity vector field. In 2010 International Conference on Electrical and Control Engineering. Wuhan, China, 25 June 2010, pp. 5022-5026;

[27] Guanglei, M.; Jinlong, G.; Fengqin, S.; Feng, T. UAV real-time path planning using dynamic RCS based on velocity vector field. In The 26th Chinese Control and Decision Conference (2014 CCDC). Changsha, China, 31 May- 2 June 2014, pp. 1376-1380”.

Besides, some key parameters we used such as V0, V1, VF, ∆V and ∆P were further explained in Lines 119-121 (caption of Figure 2). “V0, V1, VF, ∆V and ∆P are vector values, which contains both numerical values and directions. The horizontal speed of the UAV was synthesized by VF and ∆V”. and Lines 130-131 for Equations 1 and 2.

In addition, we agree with you that this study focused on more tests than development of a method, so we adjusted the title as “Precision Landing Test and Simulation of the Agricultural UAV on Apron”. Besides testing three UAV landing scenarios, we also simulated the landing in a poultry house (see details in newly added sections 2.6 and 3.4).

  1. Furthermore, once the title and abstract state about agriculture UAV, it will be supposed to show in the paper a UAV with some particular agricultural applications (that made it names agriculture UAV). However, neither simulation nor experiment related to such an UAV is showed. The figures for landing accuracy in subsection 3.3 are not enough to demonstrate the feasibility and effectiveness of the proposed method.

Reply:  Thank you for the comment/suggestions. Our plan is using this UAV for agricultural data collection such as crop lands, greenhouses, and poultry houses. We added some ongoing works about the simulation of UAV landing in a poultry house such as two new sections (section 2.6 and section 3.4):

Reply:  Thank you for the comment/suggestions. Our plan is using this UAV for agricultural data collection such as crop lands, greenhouses, and poultry houses. We added some ongoing works about the simulation of UAV landing in a poultry house such as two new sections (section 2.6 and section 3.4):

"2.6 Poultry house UAV landing simulation: In poultry or livestock houses, environmental quality monitoring and animal imaging are important tasks for evaluating if animals have a good production environment and how their health and welfare are affected in commercial production system [30, 31]. Traditional methods (e.g., installing sensors or camera in fixed place) have limitations in selecting representative data collection locations due to the large size of the poultry house (e.g., 150 m length and 30 m wide) [32]. Therefore, equipping a UAV with different air quality sensors and cameras will promote the precision sensing of poultry housing environment and animal health/welfare in the future.

In this study, we simulated the UAV landing control by using the physical environments of a commercial poultry house as a model to evaluate how the parameters of an apron can be designed for a precision landing control. The simulating house was measured 135 L × 15 W × 3 H m. We simulated that the UAV flew a round trip hourly and return back to a fixed location apron. In the simulation model, the image label was assumed as a large icon (35 cm×35 cm) and a small icon (4 cm×4 cm). The apron was set as 1 m high above floor in the end of the poultry house. Running speed of the UAV was set as 2 km/h, 3 km/h, and 4 km/h, respectively, to evaluate the effect of running speed on landing accuracy; the UAV running height was set as 2 m; the apron was assumed as a circle iron metal (1 m in diameter) at the height of 1m above floor in the end of the house. The flying posture of the UAV could be adjusted automatically according to detected images label and θ (e.g., θ need to be ≥ 45° for control) based on the method in Eqs (1) and (6).

3.4. Simulation results for the poultry house: We simulated the UAV landing control by using a real poultry house as a model to evaluate the the effect of running speed on landing accuracy during applying the UAV in poultry house (e.g., monitor indoor environmental quality and animal health, welfare, or behaviors). It was estimated that a round trip of UAV would be 270 m in the poultry house. As the poultry house was 3 m high only (from ground to ceiling), so the flight height was recommended as 1.5-2.5 m for the UAV considering the facilities restrictions. Table 5 shows the effect of UAV running speed on landing accuracy. The UAV running at a lower speed had a higher landing accuracy. In animal houses, a lower running speed is recommended considering the complex facilities and environment as compared to open field. The accuracy information provides a basis for optimize apron design in size to reduce the space use in the poultry house. This study is ongoing for studying the effect of the UAV weight on landing control efficiency to quantify how many sensors and cameras an UAV may carry in the future." 

  1. Generally, the paper is written not very well in its previous version and it still seems to me that this version is not significantly improved. The way the authors present their methods, contributions, and demonstrations are all not clear and not convinced.

Reply: Thank you for the suggestions to improve the writing. We added or modified a number of places in the writing:

  • The writing of scientific questions and objectives of this study was refined (lines 60-68)
  • The materials and methods were strengthened by adding the simulation in a poultry house (section 2.6 Poultry house UAV landing simulation);
  • The results and discussion were strengthened by including the simulation results by using a poultry house as a model (section 3.4 Simulation results for the poultry house);
  • The abstract was refined by clearly present research objectives and results (lines 16-31).
  • Conclusions were refined by list research summary in more clear way (lines 330-349).
  • Five new references were added to strengthen the section of Materials and Methods (e., References #26, #27, # 30, #31, and #32).